# Investigation of the Pathogenesis of Lumpy Skin Disease Virus in Indigenous Cattle in Kazakhstan

**DOI:** 10.3390/pathogens14060577

**Published:** 2025-06-10

**Authors:** Lespek Kutumbetov, Ainur Ragatova, Moldir Azanbekova, Balzhan Myrzakhmetova, Nurbek Aldayarov, Kuandyk Zhugunissov, Yergali Abduraimov, Raikhan Nissanova, Asylay Sarzhigitova, Nazerke Kemalova, Arman Issimov

**Affiliations:** 1Research Institute for Biological Safety Problems, Gvardeiskiy 080423, Kazakhstan; lespek.k@gmail.com (L.K.); azanbekova.m@biosafety.kz (M.A.); b.myrzakhmetova@biosafety.kz (B.M.); kuandyk_83@mail.ru (K.Z.); 2Department of Veterinary Medicine, A. Baitursynov Kostanay Regional University, Kostanay 110000, Kazakhstan; ragatova.ainur85@gmail.com; 3Department of Biology, Kyrgyz-Turkish Manas University, Bishkek 720042, Kyrgyzstan; nurbek.aldayarov@manas.edu.kg; 4QazBioPharm National Holding, Astana 010000, Kazakhstan; abduraimov_72@mail.ru; 5Department of Virology, Kazakh Scientific Research Veterinary Institute, Almaty 050016, Kazakhstan; raihan.nisanova@gmail.com; 6Department of Biology, K. Zhubanov Aktobe Regional University, Aktobe 030000, Kazakhstan; asilay_94.94@mail.ru

**Keywords:** lumpy skin disease, LSDV, experimental inoculation, Kazakhstan

## Abstract

This study investigates the virulence properties and pathogenetic characteristics of the Kazakhstani strain of LSDV (LSDV KZ-Kostanay-2018) in indigenous cattle under controlled conditions. Twelve non-breed cattle were inoculated intradermally and monitored for clinical, pathological, and immunological responses. Clinical signs, including fever, skin nodules, and lymphadenopathy, emerged as early as day 5 post-infection (pi), with peak severity observed between days 11 and 14. Rapid seroconversion was observed, with 100% of animals showing virus-neutralizing antibodies by day 13. Pathological findings revealed extensive necrosis, thrombosis, and edema, with pronounced damage in the spleen, lungs, and lymph nodes. Histological analyses identified widespread destructive changes in the dermis and systemic tissues, consistent with highly aggressive disease progression. Viral genome and replication were confirmed in blood, skin nodules, and lymph nodes, with peak viral loads between days 11 and 14 pi. These results align with findings in Russian cattle infected with the Saratov/2017 strain but demonstrate more rapid symptom onset and severe pathology, suggesting strain-specific virulence. These findings contribute to a deeper understanding of LSDV pathogenesis and underscore the importance of regional adaptations in disease management.

## 1. Introduction

Lumpy skin disease (LSD) is an economically devastating viral disease of cattle [1]. The etiological agent, lumpy skin disease virus (LSDV), belongs to the family *Poxviridae*, genus *Capripoxvirus*. LSDV shares antigenic similarity with both sheeppox and goatpox viruses [1,2,3]. Morphologically, LSDV is identical to the goatpox virus, exhibiting a spherical shape, a double membrane, and a dense core, with a virion size ranging from 260 to 320 nm [4]. LSDV has become endemic across nearly the entire African continent and has spread to the Middle East, Central Asia, China, and several neighboring countries in Southeast Asia, including Vietnam, Cambodia, Laos, Thailand, Malaysia, Singapore, and Indonesia [5,6,7,8,9,10]. Data from Mainland China, African, Middle Eastern, and South Asian countries, where LSD have been reported, indicate that outbreaks predominantly occur between April and October. This seasonality is likely associated with the peak activity phase of blood-feeding insects [5,6,8,11,12,13,14]. 

Different virus isolates can exhibit a wide range of clinical manifestations, ranging from mild to severe, with disease progression varying from asymptomatic to severe forms [2,3]. The pathogenesis and tissue tropism of LSD virus are similar to other *Capripoxvirus* infections, such as sheeppox and goatpox, characterized by lymphadenopathy despite the absence of high viral titers in lymphoid tissues [4,11]. The skin is the primary target for viral replication, with skin lesions and the nasal mucosa containing high viral loads. In cattle infected with LSDV, viremia is less pronounced compared to that observed in sheep and goats infected with sheeppox and goatpox viruses [4,11].

Studies indicate that the pathogenesis of LSD is influenced by factors such as the animal’s age, immune status, strain virulence, and inoculation routes. The inoculation route plays a critical role in the clinical course and severity of infection. For instance, intravenous inoculation of Holstein calves did not detect the presence of the virus in the bloodstream over a 42-day period, despite clinical manifestations [4]. In contrast, intravenous inoculation of bulls aged 11–20 months resulted in viremia during the first 12 days post-infection [12]. A recent study has demonstrated that the virus can be detected in the skin, testicles, trachea, lungs, and muscle tissues associated with the basal region of a skin lesion of infected animals [15]. Moreover, others showed the presence of the virus in the bloodstream for up to 4 days and in the skin nodules for up to 33 days following the onset of fever and generalized skin lesions [16], in the nasal cavity for up to 10 days, in the oral cavity for up to 8 days [17], and in bull semen for up to 159 days [18]. Recent studies have further confirmed the presence of LSDV in various biological matrices under both experimental and field conditions. For instance, Gupta, Pravalika [19], detected viral DNA in 6.89% of blood samples and 6.38% of nasal swabs from naturally infected cattle. Bedeković and Šimić [20] reported LSDV genome detection in nasal swabs from vaccinated cattle between 10 and 21 days post-inoculation. Furthermore, conjunctival (ocular) swabs have been shown to yield positive LSDV PCR results between 6 and 14 days post-infection [17]. These findings underscore the importance of using blood, nasal, ocular, and oral swabs as standard matrices for understanding LSDV pathogenesis and optimizing diagnostic protocols.

The most common clinical signs of the disease include skin nodules ranging from 0.5 to 5 cm in diameter, which may also appear on the mucous membranes of the nose, mouth, eyes, and genitalia. These nodules may regress into firm masses or develop into deep ulcers. The highest concentration of papules is typically observed in hairless regions such as the perineum, udder, inner ear, muzzle, and eyelids [2,21]. The host’s susceptibility to LSD is influenced by multiple factors, including the virus’s virulence, as well as the host’s immune status, age, and breed [22]. The incubation period ranges from 2 to 5 weeks under natural conditions, while experimental inoculation results in an incubation period of 1–3 weeks [23,24]. According to veterinary guidelines, the incubation period in experimentally infected animals is typically 4–7 days, whereas in naturally infected animals, it may extend up to 5 weeks [25].

Furthermore, studies have demonstrated that LSDV is capable of altering its biological and molecular properties. Variations in the data regarding the pathogenesis of LSD suggest a high degree of phenotypic variability, driven by exposure to diverse physical factors [26]. Parameters such as virulence and transmissibility are not constant, as they are influenced by unique environmental conditions, host-specific characteristics, infectious dose, and the route of inoculation [22].

The collapse of the Soviet Union weakened disease control systems, contributing to the emergence of transboundary diseases such as lumpy skin disease, first reported in Kazakhstan’s Atyrau region in 2016 [27,28]. The outbreak in an affected herd was characterized by the development of cutaneous nodules, deterioration in the general health of the animals, and upper respiratory tract involvement. During the outbreak, LSDV was isolated, its cultural properties were studied, and the strain was deposited in the Microorganism Collection Laboratory of the Research Institute for Biological Safety Problems (RIBSP) [29]. Two years later, in 2018, an additional strain of LSDV was isolated and identified in the Kostanay region [30]. However, the virulence characteristics of this newly isolated strain remained uninvestigated, providing the basis for reproducing the infection under controlled experimental conditions.

This study aims to investigate the virulence properties of the lumpy skin disease virus strain specific to Kazakhstan, with a focus on its potential pathogenetic characteristics under local environmental conditions in indigenous cattle breeds. 

## 2. Materials and Methods

### 2.1. Virus

The “LSDV KZ-Kostanay-2018” strain was used for animal infection. This strain was obtained from the Microorganism Collection Laboratory of the RIBSP. The virus was passaged three times on primary lamb testis (LT) cell culture, confirmed genetically, and registered in the GenBank under accession number MT992618.1. The titer was determined to be 10^5.50^ TCID_50_/cm^3^. A phylogenetic analysis based on the complete genome sequence of the isolated strain (GenBank accession no. MT992618.1) revealed its close genetic relationship to classical field strains of LSDV [30]. The isolate clustered with the Neethling LW 1959 reference strain and other classical LSDV strains previously detected in Russia, Turkey, and Israel. In contrast, it was clearly distinct from recombinant (chimeric) genotype 2.5 strains. These findings indicate that the strain belongs to the classical LSDV lineage and does not exhibit features characteristic of vaccine-derived or recombinant strains.

### 2.2. Animal Ethics

The study was conducted according to the guidelines of the Declaration of Helsinki and approved by the Institutional Review Board and Ethics Committee of the Research Institute for Biological Safety Problems of the Science Committee of the Ministry of Higher Education and Science of the Republic of Kazakhstan (permit number: 0701/021).

### 2.3. Experimental Animals

The manifestation and progression of LSD, as well as the pathogenicity of the virus, were studied in 12 indigenous cattle. Prior to the experiments, they were quarantined in an ABSL-3 facility for one month and tested for the presence of capripox antibodies using a serum neutralization assay. Water and feed were provided at will. LSDV-infected animals were monitored daily for clinical manifestations of LSD. 

### 2.4. Experimental Design

In this study, twelve clinically healthy heifers aged 12–15 months were randomly allocated into two groups (n = 6 per group). The first group (n = 6) was inoculated intradermally with a total of 2 mL of LSDV at a dose of 4 × 10^3^ TCID_50_/mL into a shaved area of the neck, approximately at the level of the third cervical vertebra, while the second group (n = 6) served as controls. Using the “envelope method”, inoculation was performed at four peripheral points and one central point (five sites in total), with 0.4 mL of virus suspension injected into each site. The animals were physically separated in isolated pens under identical husbandry conditions to prevent cross-contamination.

Samples (nasal, ocular swabs, and whole blood) were collected on days 3, 5, 7, 9, 11, 13, 15, 21, and 28 post-inoculation (pi) for PCR analysis. Nasal and ocular swabs were collected using sterile synthetic swabs (Copan Diagnostics, Murrieta, IA, USA), each placed in 1 mL of phosphate-buffered saline (PBS). Whole blood samples were collected into EDTA-containing vacutainer tubes (purple top) for DNA extraction and PCR testing. Similarly, blood serum samples were also drawn into serum-separating tubes (SST, yellow top) without anticoagulant for serum harvesting. The obtained sera were used for ELISA and serum neutralization tests on days 3, 5, 7, 9, 11, 13, 15, 21, and 28 pi. Animals were sacrificed on days 7, 9, 11, 15, 21, and 28 post exposure. At each time point, one animal from each group (infected and control) was sacrificed to assess histopathological alterations in skin nodules and internal organs. This approach allowed for a systematic evaluation of changes over time. Additionally, skin nodules, as well as prescapular and submandibular lymph nodes, were examined for the presence of the virus and its genomic confirmation at the aforementioned intervals. The severity of clinical response was evaluated based on clinical signs, including their diversity and intensity (Figure 1).

### 2.5. PCR

DNA extraction was performed using the “DNeasy^®^ Blood & Tissue Kit (250)” (QIAGEN, Stockach, MD, USA) following the manufacturer’s instructions. Prior to extraction, the samples were subjected to cryodestruction. PCR assays for the detection of genetic determinants of capripoxviruses were conducted using the Tag Polymerase SYBR Green kit (Sigma-Aldrich, St. Louis, MO, USA) with genus-specific primers, amplifying a 192-base pair product. The primers used were purchased from Thermo Fisher Scientific (Waltham, MA, USA) and were as follows: forward 5′-TCC-GAG-CTC-TTT-CCT-GAT-TTT-TCT-TAC-TAT-3′ and reverse 5′-TAT-GGT-ACC-TAA-ATT-ATA-TAC-GTA-AAT-AAC-3′ [31]. PCR amplifications were performed using a Bio-Rad CFX96 Real-Time PCR Detection System (Bio-Rad, Hercules, CA, USA) under the following thermal cycling conditions: initial denaturation at 95 °C for 45 s, followed by 35 cycles of 50 °C for 50 s, 72 °C for 60 s, 72 °C for 2 min. 

### 2.6. ELISA

To detect LSDV-specific antibodies a commercial ELISA kit (ID Screen^®^ Capripox Double Antigen Multi-species, ID Vet, Grabels, France) was utilized following the manufacturer’s instructions.

### 2.7. Serum Neutralization Test 

To assess neutralizing antibody titers against LSDV, a serum neutralization test was conducted following the guidelines outlined in the Terrestrial Manual for Diagnostic Tests and Vaccines [31]. Virus-neutralizing antibody (VNA) titers were determined against 100 TCID_50_ of the LSDV field strain (LSD KZ-Kostanay-2018). Positive control antisera (obtained from an animal vaccinated with a homologous LSDV vaccine) and negative control antisera were included in the assay. Briefly, sera were serially diluted (1:2 to 1:128) in 96-well flat-bottom microtiter plates, followed by the addition of 100 TCID_50_ of the LSDV field isolation to each well. The plates were incubated at 37 °C for 1 h and subsequently at 4 °C overnight. Following incubation, 50 μL of lamb kidney (LK) cell suspension (2 × 10^5^ cells/mL) was added to each well, and the plates were incubated at 37 °C in the presence of 5% CO_2_ for 4–7 days. The plates were then examined for LSDV-induced cytopathic effects (CPE). Neutralization titers were expressed as the reciprocal of the serum dilution at which 50% neutralization was achieved.

### 2.8. Evaluation of the Clinical Reaction

The clinical response for each animal was assessed based on a comprehensive clinical examination, following the methodology described by Carn and Kitching [32]. Scores ranging from 7 to 10 represented generalized disease with varying degrees of severity, while scores between 1 and 6 reflected the extent of the localized reaction at the inoculation site and the associated lymphadenopathy.

### 2.9. Histopathology

Tissue samples were collected from affected skin (nodules), lymph nodes, spleen, lungs, liver, affected muscles, rumen, kidneys, and heart. Specimens were obtained aseptically by excising 2–4 g of biopsy material. The tissues were placed in sterile disposable containers containing 10% formalin for preparation for histopathology. Histological sections of 5 µm thickness were prepared from paraffin-embedded blocks and were stained with hematoxylin and eosin (H&E), following standard operational procedures of the BSL-3 laboratory of the RIBSP. 

### 2.10. Virus Isolation

Virus isolation was performed following the standard operating procedures of the BSL-3 laboratory at RIBSP, in accordance with WOAH [31]. In brief, 1 mL supernatant was inoculated onto lamb testicular cells monolayers in 25 cm^2^ cell culture flasks and incubated at 37 °C for 1 h. After incubation, the cells were rinsed with PBS and overlaid with Glasgow’s MEM supplemented with 2% fetal calf serum (Atlanta Biologicals, Lawrenceville, CA, USA), 5, 10 U/mL penicillin and 10 µg/mL streptomycin sulfate (HEPES). The cell monolayer was observed daily for the presence of characteristic cytopathic effects. If no CPE was detected, the cell culture underwent three freeze-thaw cycles, and two to three blind passages were performed. The culture media were stored at −80 °C until further analysis. Flasks exhibiting CPE were subsequently tested by PCR to confirm that the observed CPE was developed by LSDV.

## 3. Results

### 3.1. Clinical Manifestations and Pathogenicity Evaluation of the Virus

The virus pathogenicity was determined based on quantitative and qualitative parameters of clinical manifestations and evaluated by the onset, intensity, and severity of disease progression.

In the experimental group, a systemic reaction to the virus administration was observed on day 4 pi and characterized by hyperthermia. On day 5 pi, body temperature elevated to 41.0 °C, followed by a fever persisting in the range of 39.5–40.0 °C over the subsequent two days (Figure 2a,b). Animals exhibiting clinical signs demonstrated by pronounced swelling at the site of virus inoculation by the second and third days of hyperthermia (day 7 pi), measuring approximately 7 cm in length, 1.5 cm in width, and 1.5 cm in thickness (Figure 3A). Additionally, swellings of the right forelimb, followed by mild lameness, were observed in one animal on day 7 pi (Figure 3B). The same animal also exhibited conjunctivitis with excessive ocular discharge. On the fourth day of clinical manifestation (day 8 pi), multiple skin lesions (4–5 per area) appeared on the chest and left side of the neck in 5 animals.

The condition of animals in the experimental group continued to deteriorate, as evidenced by increased lethargy, emaciation, enlargement of superficial lymph nodes, and reluctance to move, feed, and drink (Figure 3C). The animals progressively lost weight, spent most of their time in a recumbent position (Figure 3D), and exhibited swelling of the forelimb caused by edema. Nasal erosions were observed, and the number of cutaneous nodules increased (Figure 3E). Among the six cattle inoculated intradermally with LSDV, clinical signs of the disease were observed in all animals: hyperthermia in four, primary skin nodules in all six, and secondary generalized nodules in three. The clinical assessment of disease severity indicated an average symptom severity score of 10 (Figure 2b). The most pronounced clinical manifestations, indicative of intensive disease progression, were observed between days 12 and 21 pi.

Animals exhibiting clinical signs demonstrated by pronounced swelling at the site of virus inoculation by the second and third days of hyperthermia (day 7 pi), measuring approximately 7 cm in length, 1.5 cm in width, and 1.5 cm in thickness (Figure 3A). Additionally, swellings of the right forelimb, followed by mild lameness, were observed in one animal on day 7 pi (Figure 3B). The same animal also exhibited conjunctivitis with excessive ocular discharge (Figure 3C). On the fourth day of clinical manifestation (day 8 pi), multiple skin lesions (4–5 per area) appeared on the chest and left side of the neck in 5 animals.

The condition of animals in the experimental group continued to deteriorate, as evidenced by increased lethargy, emaciation, enlargement of superficial lymph nodes, and reluctance to move, feed, and drink. The animals progressively lost weight and spent most of their time in a recumbent position (Figure 3D). Nasal erosions were observed (Figure 3C), and the number of cutaneous nodules increased (Figure 3C). Among the six cattle inoculated intradermally with LSDV; clinical signs of the disease were observed in all animals: hyperthermia in four, primary skin nodules in all six, and secondary generalized nodules in three. The clinical assessment of disease severity indicated an average symptom severity score of 10 (Figure 2b). The most pronounced clinical manifestations, indicative of intensive disease progression, were observed between days 12 and 21 pi.

### 3.2. Serum Neutralization Test

The evaluation of VNA titers using the SNT at various observation periods revealed the dynamics of the animals’ immune response to viral inoculation (Table 1). On day 7 pi, a positive response was detected in 4 animals at a titer of 1:2, indicating the initiation of an immune response. By day 11 pi, a positive response was observed in 100% of the animals at a titer of 1:4, reflecting a significant enhancement in immune activity. On day 13 pi, all animals also demonstrated a 100% positive response at a titer of 1:8, confirming the rapid development of the immune response. From days 15–17 pi, a stable positive response was observed in all animals with high titers, particularly at a titer of 1:8 and 1:16, where 2 animals retained a positive response by day 17 pi. Thus, this progression underscores the increasing immune resilience against the virus.

### 3.3. ELISA

The LSDV antibody levels were elevated in the peripheral blood of cattle during the observational period (Figure 4). On days 3 and 5 pi, antibody levels in all animals remained below the threshold value (30%), indicating the absence of a significant immune response. By day 7, antibody titers in several animals (n = 3) reached approximately 40%, marking the initial stage of seroconversion. By day 11, most animals (n = 5) exhibited antibody levels exceeding the threshold value (S/P ≥ 30%), with levels reaching 65% or higher in a substantial proportion of animals, suggesting immune system activation and the production of specific antibodies against LSDV. Starting from day 13 pi, antibody levels steadily increased, reaching peak values by day 21 (Figure 5). At this stage, all animals exhibited pronounced seroconversion, with antibody levels exceeding 85%, indicating a robust immune response. These findings confirmed the anticipated dynamics of the immune response to LSDV in cattle, characterized by a significant rise in antibody levels during the second week post-infection and peaking on day 13.

### 3.4. PCR

Detection of the LSDV genome specificity was performed through the identification and species differentiation of capripoxviruses using real-time PCR. DNA was extracted from virus-containing material, and the results of genome detection are presented in Table 2.

LSDV DNA was detected in whole blood and skin nodule samples starting on day 7 post-infection (Table 2). Given the presence of clinical signs characteristic of LSD in the affected calves, 20% organ tissue suspensions were utilized for virological studies. The investigation involved isolating the virus from pathological material using cell cultures, which resulted in the identification of an agent causing CPE in the cultured cells. Analysis of the cytopathic agent using PCR assay confirmed that this etiological agent is LSDV. Detailed results of the study are presented in Table 2.

### 3.5. Virus Isolation

The highest level of virus shedding was observed in whole blood, starting on day 7 pi, with a peak titer of 4.5 log TCID_50_/mL on day 11 pi, followed by a gradual decline. In nasal and ocular swabs, the virus was detected on day 11 pi, with maximum titers 2 and 2.75 log TCID50/mL observed on days 13 and 15, respectively, after which the virus titer also declined. In skin nodules, LSDV was detected from day 7 pi, reaching peak titers 4.75 and 5.0 log TCID_50_/mL on days 11 and 13, respectively. By day 21 pi, no virus shedding was observed from skin nodules, indicating the onset of immune system-mediated infection control. In lymph nodes, the virus was detected from day 7 pi, with titers (2.0 log TCID_50_/mL) peaking on day 14 pi. Thus, peak virus shedding was observed between days 11 and 14 pi, reflecting the active phase of the infectious process. Subsequently, a gradual decline in viral titers was observed, indicating the initiation of the host’s immune response.

### 3.6. Postmortem Examination 

During necropsy on the 11th, 15th, and 21st days post-infection, pathological changes were observed in the internal organs (lungs, intestines, trachea, spleen, subscapular lymph nodes, submandibular lymph nodes, and skin), characteristic of LSD (Figure 6). Nodules, edema, congestion, and vascular thrombosis were observed across all examined organs and tissue systems (Figure 6a–c).

### 3.7. Histopathological Alterations in the Skin

Microscopic examination of all collected samples revealed various stages of organ structural damage. Alterations were characterized by multiple necrotic areas and inflammatory infiltration of varying severity. Destructive changes in the skin in some regions were confined to the dermal layer, while the epidermal layer remained relatively intact.

All samples exhibited profound destructive changes in the dermis. Completely necrotized areas contained pronounced vacuolar degenerative fibrotic structures composed of small nodules of varying shapes (Figure 7a). These nodules included eosinophilic homogeneous coagulative masses as well as small clusters of cellular debris described as “cellular detritus”. Such formations were surrounded by fibrotic tissue, with their marginal zones consisting of smaller vacuolar degenerative structures, dense necrotic masses, damaged blood vessels of various calibers, and significant infiltration by polymorphonuclear and mononuclear inflammatory cells. Multiple solitary apoptotic bodies and petechial hemorrhages were identified. Some nodules also exhibited streak-like hemorrhages (Figure 7b). In other areas of the dermis, the microstructures of hair follicles were predominantly destroyed (vacated), with only a few remnants preserved. Mild sebaceous gland hyperplasia was noted (Figure 7c), along with vasculitis of varying severity (Figure 7d), perivascular inflammation, and perivascular necrosis (Figure 7e). In some arterial vessels, hyperplasia of smooth muscle cells resulted in luminal narrowing. The affected vessels were surrounded by inflammatory cells, predominantly macrophages containing intracytoplasmic viral inclusion bodies (Figure 7f). Coagulative hyaline necrosis and calcification of muscle fibers were observed (Figure 7g). Intermuscular spaces were enlarged and infiltrated with focal accumulations of inflammatory cells (Figure 7h).

As previously mentioned, the epidermis in certain areas retained partial microstructural integrity. However, these regions were relatively atrophied, particularly the prickle cell layer of the epidermis. This layer exhibited swollen cells, intercellular edema (spongiosis), and isolated elements of hydropic degeneration in the form of intraepithelial microvesicles of varying sizes. The stratum corneum showed hyperplastic acanthosis (Figure 7i). In other areas, the epidermis was significantly thinned, with only basal and granular layer keratinocytes and the stratum corneum remaining intact. The pathological process was largely accompanied by hyperkeratosis (Figure 7i) and parakeratosis (Figure 7j). The affected epidermis distinctly exhibited keratinocytes containing characteristic viral inclusion bodies of varying sizes and shapes (Figure 7k). Hyperplasia and necrosis of the basal layer keratinocytes were also observed, while the remaining epidermal layers were absent (Figure 7l). Additionally, numerous individual or clustered apoptotic bodies and cellular debris were identified within the necrotic hyaline masses (Figure 7l). In severely affected regions, the epidermis was completely atrophied or ruptured.

### 3.8. Histopathological Alterations in Other Organs

*Lymph Nodes.* Significant destructive changes were observed, characterized by vacuolar degeneration with fibrous structures, hyperemia, hemorrhages, subcapsular sinus edema, profound vascular wall alterations, and multifocal areas of necrotic masses in the medulla and intermediate sinuses of the lymph node (Figure 8a). Numerous smaller vacuolar degenerations were widely distributed in the paracortical and follicular zones of the cortex (Figure 8b). Atrophy of individual lymphoid follicles and substantial accumulation of macrophages in the medullary sinus were noted. Across all regions of the lymph node, a high number of apoptotic bodies were present.

*Spleen.* Numerous vacuolar degenerations were diffusely scattered, particularly in the subcapsular zone (Figure 8c) and within the lymphoid follicles of the white pulp (Figure 8d). Pronounced atrophy of the lymphoid tissue in the white pulp was observed, along with isolated necrotic masses surrounded by macrophages and other immunocompetent cells, as well as destructive alterations in the walls of arterioles.

*Lungs.* A significant portion of the parenchyma exhibited fibro-purulent necrosis (Figure 8e), accompanied by remnants of cellular debris and numerous vacuolar structures of varying sizes and shapes. Pronounced signs of congestion, extensive hemorrhages, vasculitis of varying severity, and focal infiltration by inflammatory cells were observed. The lumens of preserved alveoli were filled with alveolar macrophages, some of which contained characteristic eosinophilic intracytoplasmic viral inclusions (Figure 8f).

*Muscle Tissue.* Observations revealed atrophy and necrosis of muscle fibers (Figure 8g,h), accompanied by an increase in the volume of intermuscular spaces, regenerative fibroblast proliferation, and an increase in connective fibers. Diffuse infiltration of inflammatory cells was noted within the intermuscular spaces (Figure 8h).

*Rumen.* Destructive alterations were observed in the stratified squamous epithelial lining, characterized by vacuolar degeneration of squamous epithelial cells (Figure 8i,j), and in the muscular layer, evidenced by necrosis and loosening of muscle fibers (Figure 8j). Pathological changes in the vascular walls were also noted, accompanied by inflammatory cell infiltration.

*Kidneys.* Severe destructive changes were identified throughout the renal parenchyma, including vacuolar degeneration, coagulative necrosis in renal corpuscles and tubules (Figure 8k,l), vasculitis, and localized infiltration by inflammatory cells, predominantly macrophages.

*Liver.* Hepatocytes (Figure 8m,n) and the central vein of the hepatic lobules (Figure 8m) exhibited necrosis of varying degrees. Minimal focal and widespread diffuse infiltration by inflammatory cells was observed across all hepatic zones (Figure 8n).

*Heart.* Necrosis and disintegration of myocardial cardiomyocytes were observed (Figure 8o,p), along with diffuse infiltration of inflammatory cells within the intermuscular spaces. Characteristic vacuolar degenerations associated with this condition were also identified (Figure 8p). 

## 4. Discussion

This study investigated the clinical progression, virological dynamics, and immune responses in cattle experimentally infected with a field strain of lumpy skin disease virus (LSDV) under controlled conditions. All inoculated animals developed primary nodules at the inoculation site, accompanied by varying degrees of systemic signs, such as fever, generalized nodules, conjunctivitis, and limb swelling. These observations are consistent with previous descriptions of LSDV pathogenesis in experimental settings [33,34].

Our study revealed that the Kazakhstani strain induces symptom onset as early as day 5 post-infection. Quantitative assessment of viral load in cell cultures demonstrated that the accumulation of the pathogen in skin nodules, blood, and lymph nodes peaks between days 11 and 14 post-infection. Virus-neutralizing antibodies were detected from day 7 post-infection, with titers increasing steadily through day 21. This serological response indicates early and sustained humoral activation following the LSDV challenge. The antibody kinetics observed align with prior reports, where neutralizing antibodies typically become detectable between 7 and 14 days post-infection, supporting the importance of timely immunological surveillance and vaccination [35,36]. It is important to note that the immune response to the LSD KZ-Kostanay-2018 strain was characterized by rapid seroconversion, with 100% of animals achieving detectable virus-neutralizing antibodies by day 13 post-infection. While this observation appears earlier than in previous reports from Russia [37] and India [38], direct comparison is difficult due to differences in animal breeds, experimental designs, and environmental conditions. Therefore, the faster antibody response in our study may reflect strain-host interactions specific to the local context. 

The pathological changes observed in affected tissues are also noteworthy. Similar to the findings reported by Flannery, Shih [39], the Kazakhstani LSD KZ-Kostanay-2018 strain exhibits necrosis, thrombosis, and edema in internal organs. However, in this case, the severity of damage, particularly in the spleen and lungs, was more pronounced, potentially indicating a unique virulence profile of the virus. The LSDV strain used in this study was a locally isolated field strain obtained during a recent outbreak in the region [30]. This strain was selected to ensure epidemiological relevance, as it reflects the genetic and pathogenic characteristics of viruses currently circulating in the local cattle population.

Histological examination of animals experimentally infected with the Kazakhstani strain of LSDV revealed pathological changes at the cellular level in all tissue samples. Significant systemic alterations were observed throughout the body, including extensive destructive lesions in the dermis. Our findings align with those reported by Badr, Noreldin [40] and Douglass, Munyanduki [41], who described pronounced destructive alterations in histological sections of affected organs. These alterations were characterized by vacuolar degeneration with fibrotic structures, hyperemia, hemorrhages, subcapsular sinus edema, as well as multiple necrotic areas and inflammatory infiltration of varying intensity. 

There are several limitations to this experimental study that should be acknowledged. Firstly, the most important constraint was decreasing sample size over time due to scheduled sacrifices for histopathological and virological assessments. This approach, while essential for detailed temporal analyses, limits the statistical power and the ability to assess individual variations in disease progression and immune responses. Although the number of animals was limited, individual variability in clinical response to LSDV infection was evident. Notably, one animal developed forelimb swelling and conjunctivitis by day 7 post-infection. Secondly, while the study confirmed the presence of the virus and its replication in tissues, in-depth molecular analyses to identify specific genetic determinants of virulence or tissue tropism were not performed. This could have provided insights into the mechanisms underlying the observed pathogenesis. Finally, it is important to acknowledge that the experimental infection model employed in this study, specifically, intradermal inoculation with a defined viral dose at a fixed anatomical site, does not fully replicate the natural transmission routes of LSDV. In field conditions, LSDV is primarily transmitted through hematophagous arthropods such as Stomoxys calcitrans, Aedes aegypti, and Culex quinquefasciatus, or via direct contact through oral and respiratory mucosa [42,43]. Natural exposure is also characterized by variable and often lower viral loads compared to standardized experimental models [44]. These differences in route, dose, and site of viral entry can significantly influence disease kinetics, including incubation period, clinical severity, and immune response patterns [17]. While intradermal inoculation is widely used in controlled studies for its reproducibility and efficiency in inducing infection, it may elicit more synchronized or exaggerated clinical outcomes compared to the heterogenous and sometimes subclinical disease presentations in field cases [44,45]. Despite these limitations, the study successfully demonstrates key features of LSDV pathogenesis and supports the need for future investigations employing natural infection models and extended follow-up to improve the translational relevance of experimental findings.

## Figures and Tables

**Figure 1 pathogens-14-00577-f001:**
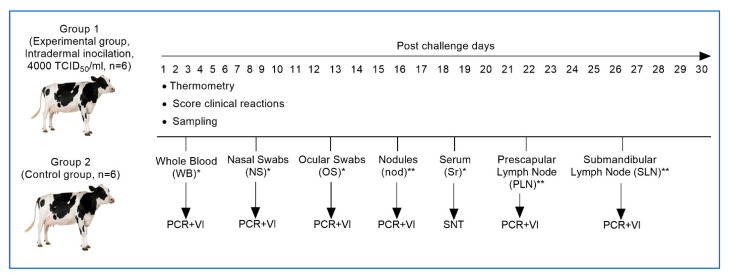
Study design for assessing the pathogenicity of the Kazakhstani (LSD KZ-Kostanay-2018) strain of lumpy skin disease virus. (*)—Samples (whole blood, nasal, and ocular swabs). (**)—Lymph nodes and nodule samples were collected on days 7, 11, 15, 21, and 28 by euthanizing one animal from each group. PCR—polymerase chain reaction; VI—virus isolation; SNT—serum neutralization test.

**Figure 2 pathogens-14-00577-f002:**
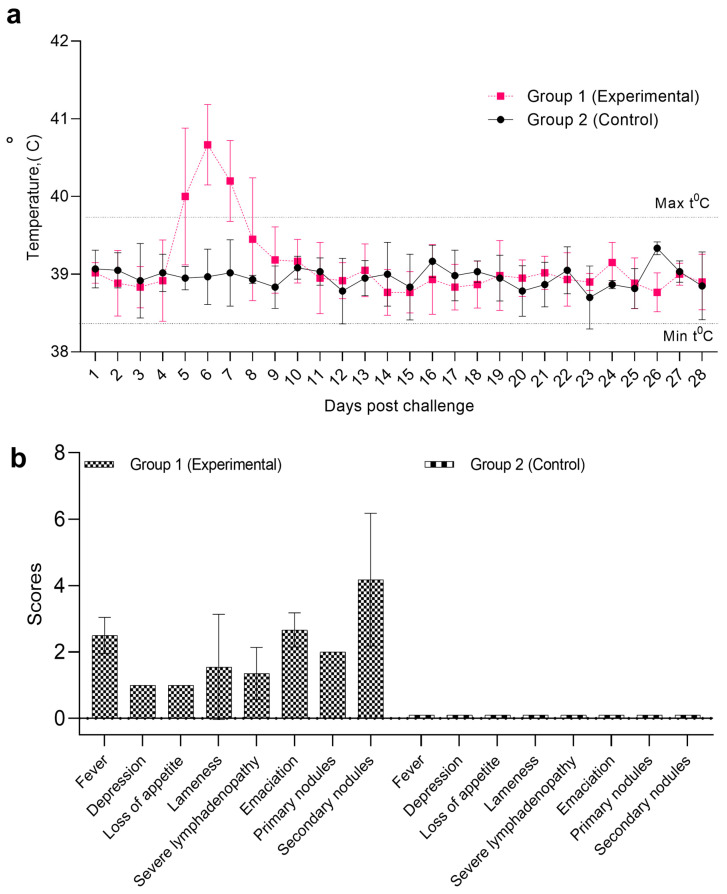
Pathogenicity of the Kazakhstani (LSD KZ-Kostanay-2018) strain of lumpy skin disease virus. (**a**)—dynamics of body temperature; (**b**)—clinical scoring LSD manifestations.

**Figure 3 pathogens-14-00577-f003:**
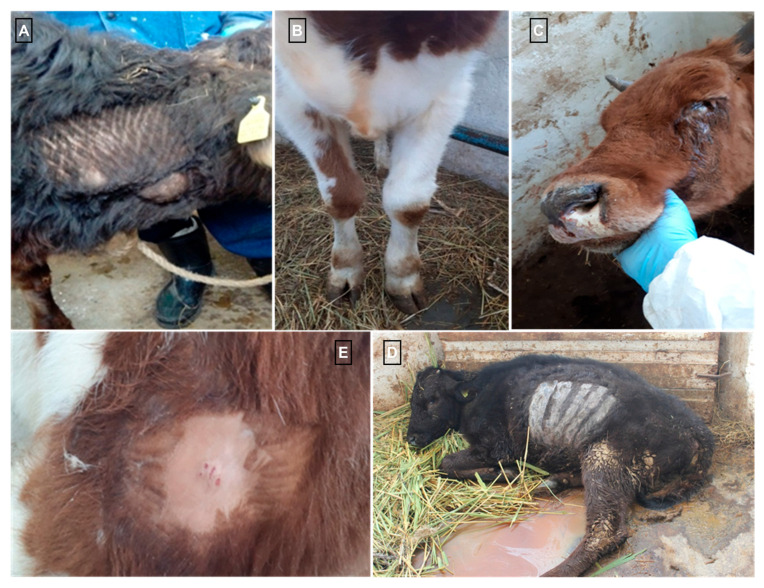
Clinical manifestations of LSD in infected animals. (**A**)—enlargement of superficial regional lymph node; (**B**)—swellings of the forelimb; (**C**)—conjunctivitis with serous discharge; (**D**)—recumbent posture of a clinically affected animal; (**E**)—skin lesions.

**Figure 4 pathogens-14-00577-f004:**
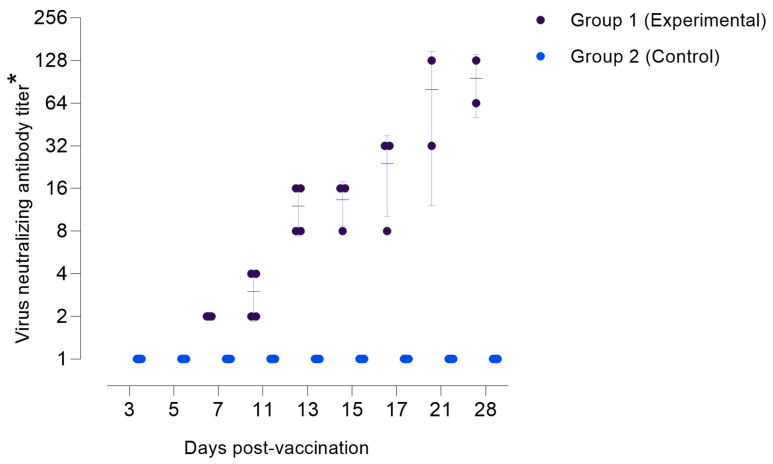
The accumulation of antibodies in the sera of experimental animals against the LSDV. (*) Titers are presented as reciprocal serum dilutions.

**Figure 5 pathogens-14-00577-f005:**
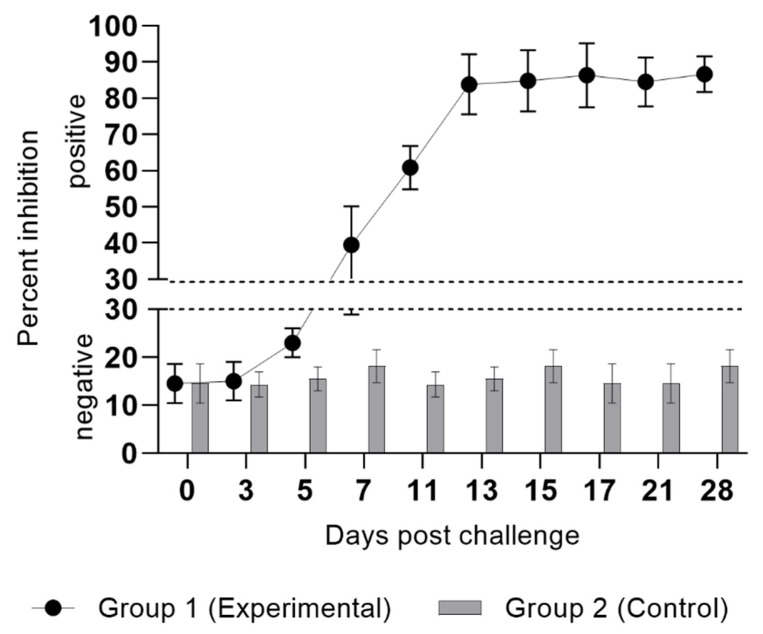
Percentage inhibition of specific antibodies in serum samples from animals following experimental infection with LSDV. Optical density values < 30 indicate negative results, whereas values > 30 indicate a positive response.

**Figure 6 pathogens-14-00577-f006:**
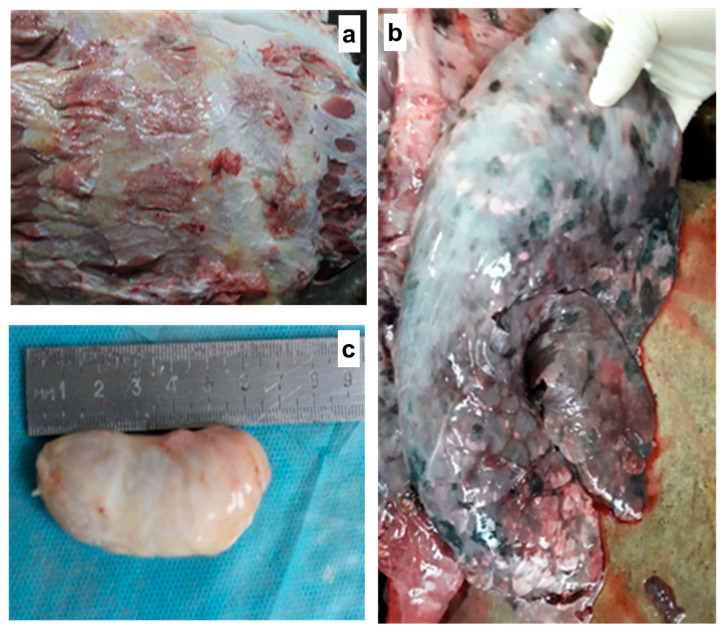
Animals necropsied during the mid and late stages of the disease exhibited necrosis and hemorrhages with serosanguinous fluid in the subcutaneous tissue (**a**). Pulmonary findings included congestion, hepatization, and multiple localized nodular lesions (**b**). The prescapular lymph node was enlarged, congested, and hemorrhagic (**c**).

**Figure 7 pathogens-14-00577-f007:**
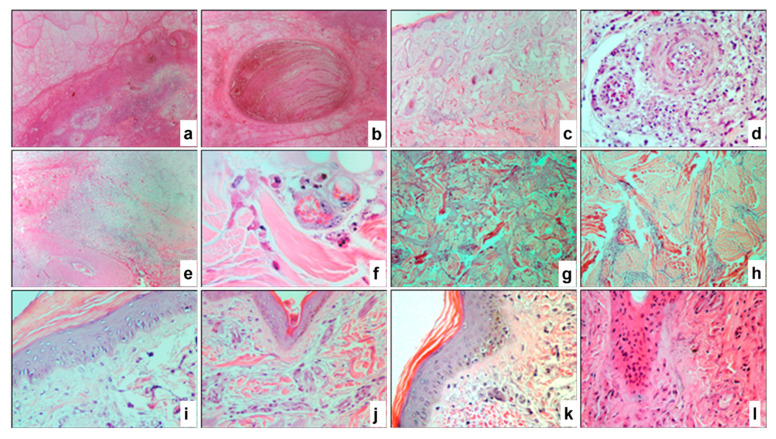
Paraffin skin sections of a cow infected with LSDV. Profound vacuolar degenerative fibrous structures formed from small nodules and peripheral focal infiltration of inflammatory cells (**a**). Necrotized oval nodule with plate-like hemorrhages (**b**). Destructive changes in the dermis with mild hyperplasia of sebaceous glands, depleted hair follicle sites, and necrosis of muscle fibers in the subcutaneous tissue (**c**). Degenerative changes in the arterial wall and cells with intracellular viral inclusion bodies (**d**), perivascular necrosis (**e**), and macrophages with characteristic viral inclusions near the affected vessel (**f**). Necrosis and calcification of muscle fibers (**g**,**h**), intermuscular space with coagulative hyaline necrosis (**g**), and focal accumulation of inflammatory cells (**h**). Acanthosis and intraepithelial microvesicles (**i**) in the spinous layer. Marked parakeratosis (**j**) and hyperkeratosis (**k**). Characteristic viral inclusions within the cytoplasm of keratinocytes (**l**), hyperplasia, and necrosis of epidermal cells (**l**). H&E staining. ×40 (**a**–**c**,**e**), ×400 (**d**,**i**–**l**), ×1000 (**f**, under immersion system), and ×100 (**g**,**h**).

**Figure 8 pathogens-14-00577-f008:**
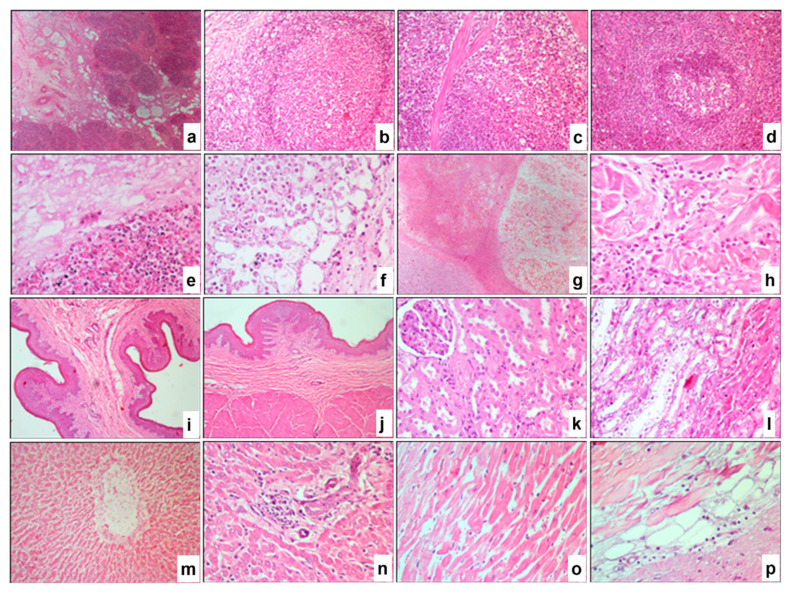
Paraffin sections of lymph nodes (**a**,**b**), spleen (**c**,**d**), lungs (**e**,**f**), affected muscles (**g**,**h**), rumen (**i**,**j**), kidneys (**k**,**l**), liver (**m**,**n**), and heart (**o**,**p**) from animals infected with LSDV. Severe destructive changes in the medullary (**a**) and cortical regions (**b**) of the lymph node. Vacuolar degeneration in the subcapsular zone of the red pulp (**c**) and the deep white pulp (**d**) of the spleen. Fibropurulent necrosis, diffuse inflammatory cell infiltration, and hemorrhage in the lungs (**e**), with alveolar macrophages containing viral inclusions (**f**). Atrophy and necrosis of muscle fibers (**g**,**h**). Vacuolar degeneration of squamous epithelial cells (**i**,**j**) and necrosis with disintegration of muscle fibers (**j**) in the rumen. Coagulative necrosis in renal corpuscles and tubules (**k**,**l**). Hepatocyte necrosis (**m**,**n**), necrosis of the central vein in hepatic lobules (**m**), and diffuse infiltration of inflammatory cells in the liver (**n**). Necrosis and disintegration of cardiomyocytes (**o**), along with characteristic vacuolar degenerations (**p**) in the myocardium. H&E. ×40 (**a**,**g**,**i**,**j**), ×200 (**b**–**d**,**m**), and ×400 (**e**,**f**,**h**,**k**,**l**,**n**–**p**).

**Table 1 pathogens-14-00577-t001:** Isolation of LSDV in lamb testis cell culture. Average viral titers (TCID_50_/mL) detected in various clinical specimens collected from LSDV-infected cattle (n = 6) at different days post-infection. Values represent mean titers for each specimen type per time point.

Samples	Sampling Time Points (Days Post-Infection) and Viral Titers (TCID_50_/mL) Obtained from Various Pathological Materials.								
	3	5	7	9	11	13	15	21	28
Whole blood	0.00	0.00	1.75	2.50	4.50	4.00	4.25	2.00	0.00
Nasal swabs	0.00	0.00	0.00	0.00	1.00	2.75	1.75	0.00	0.00
Ocular swabs	0.00	0.00	0.00	0.00	1.25	2.00	2.00	0.00	0.00
Skin nodules	n/a	n/a	3.50	4.75	4.75	n/a	5.00	0.00	0.00
Prescapular lymph nodes	n/a	n/a	1.75	1.75	1.00	n/a	1.00	0.00	0.00
Submandibular lymph nodes	n/a	n/a	1.25	1.25	1.25	n/a	1.75	0.00	0.00

«n/a»—not applicable.

**Table 2 pathogens-14-00577-t002:** Detection of antigen for LSDV using PCR.

Samples	Days Post-Infection								
	3	5	7	9	11	13	15	21	28
	Ct Value *								
Whole blood	-	-	25.02	21.14	20.02	21.01	20.17	29.50	-
Nasal swabs	-	-	-	27.88	28.25	25.89	28.87	-	-
Ocular swabs	-	-	-	25.13	28.18	28.06	-	-	-
Skin nodules	n/a	n/a	19.76	15.45	18.98	n/a	21.03	24.56	29.84
Prescapular lymph nodes	n/a	n/a	-	29.32	29.85	n/a	-	-	-
Submandibular lymph nodes	n/a	n/a	29.41	28.56	29.78	n/a	30.00	-	-

* «Ct > 30»—negative; «Ct < 30»—positive. «-»—not detected. «n/a»—not applicable.

## Data Availability

The data that support the findings of this study are available from the author, A.R., upon reasonable request.

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
