# Peer review of "Investigation of the Pathogenesis of Lumpy Skin Disease Virus in Indigenous Cattle in Kazakhstan"

_pathogens, 2025, doi:10.3390/pathogens14060577_

Round 1
Reviewer 1 Report
Comments and Suggestions for Authors
The authors present results from a pathogenesis experiment in cattle with an LSDV strain isolated in Kazakhstan. The data collected are interesting in principle, although some methodological shortcomings and missing information need to be addressed.
In my opinion, the following additional information must be included in the paper or statements in the article must be changed.
- The term “Neethling virus” should be avoided in the text; the virus is referred to as lumpy skin disease virus (LSDV) and should always be named as such.
- Lines 55-66: This section needs to be completely rewritten. Many publications in which the different matrices (blood, nasal swabs, eye swabs, mouth swabs) were also tested in comparable pathogenesis studies have not been cited and have not been taken into account in the description of the initial situation.
- Line 105: The sequence of the virus used in the study is known. Unfortunately, the phylogenetic assignment to other LSDVs from the region and worldwide is missing. A phylogenetic tree showing the relationship to classical LSDV and chimeric LSDV (genotype 2.5) strains should be created (see Breman et al 2023).
- Line 119ff: Much information on the experimental design is missing.
- How were the two groups of animals physically separated?
- How exactly was the intradermal inoculation performed? How many inoculations were performed? On one side of the neck or on both sides? How many µl were applied per inoculation site? The administration of 2 mL of inoculum cannot have been performed intradermally via one inoculation site.
- How were the nasal and ocular swabs processed? Which swabs were used? How much liquid was used to shake out the swabs? PBS or cell culture medium?
- Line 141ff: Was a Taq Polymerase SYBR Green kit used? Which company? Number of cycles, which real-time cycler?
- Line 221: Figure 2: Images c) to h) are too small. It is difficult to see anything. Graphics a) and b) should be a separate figure. The images of the animals should form a novel graphic. Consider which images should be enlarged and which can be omitted.
- Line 239 Table 1: An alternative representation should be chosen for each animal and day. To do this, the animal numbers must be listed and the SNT titers determined for each day. The data from the six control animals must also be listed there, similar to Figure 3.
- Line 262 Table 2: Use a period instead of a comma for the Ct values. What do the Ct values in bold mean? Please explain.
- Line 289: Figure 4: Make the images larger and only show the most important ones. The nodules are not easy to see and can only be guessed at.
- Line 423-25: A higher virulence for this LSDV strain is pure speculation and is not proven by the studies presented. Different experiments are difficult or impossible to compare because of different animals, conditions, and experimenters. Comparisons of the virulence of different strains require simultaneous analysis under the same conditions or large studies at the population level.
- Line 430-34: The statements on the rapid detection of neutralizing antibodies are pure speculation. Other causes are also possible here. These include, for example, the route of inoculation of the animals, the amount of virus, the age and condition of the animals used, but also the SNT method itself. Many laboratories start with a 1/5 or 1/10 dilution of the test serum. Here, too, sera from different experiments can only be reliably compared if they are analyzed in parallel in one laboratory.
Reviewer 2 Report
Comments and Suggestions for Authors
Kutumbetov et al experimentally inoculated 6 bovines with lumpy skin disease virus. Using several diagnostic methods they gave a detailed picture about the acute phase of the infection, by serology (ELISA, SNT), PCR, necropsy etc. Although the virus and the disease is known for decades, such works are necessary, as the illness is rare, most of vets never seen this in their practice. Although the work does not add any novel discovery, it is useful, as helps diagnosticians in better identification of the disease either in the beginning, in the onset and declining parts of the disease progress. For veterinaries of not endemic areas it is important.
Introduction.
This virus is called Lumpy skin disese virus, LSDV. How and why was the agent named as Nethling virus?
In the introduction chapter the authors should spend 1-2 sentences about the spread of the disease in the world. Where is this virus endemic?
lines 96-99 – meaningless empty sentence, omit it.
Matherials and methods.
The lamb testicle cells are an established cell line or a primary cell culture?
- line 114 – 12 head?? non-breed What does these mean? 12 are heads.
- were studied with a group (n - 12) of 12-15 months old indigenous cattle.
- gender of the animals (bulls, cows?)
- lines 122 -125. These two sentences should be united.
- 125 -126. One animal was sacrificed daily from both groups??? Should be cleared. The authors should not spend 6 animals as controls, they did not add too much. 1-2 control animals would have been enough to show, that they remained uninfected, the virus did not spread. Controls are for that. 2 controls should be kept untill the end of the experiment, than by ELISA, PCR their negativity should have been proven. In this way more animals would remained for the experimentally infected branch, providing possibility to follow the infection longer, or more detailed.
- lines 131 - 133. Useless sentence, should nbe omitted.
- - 141. The primers were developed by the authors, or they got from somebody else?
- line 191. Why was thawings necessery here?
- How and where the animals were kept? On the meadow, in a stable? Together with the controls or separated? How the authors prevented the virus from spreading to animals not involved in the study?
Results.
line 201 – first day of illness. post infection days should be used.
- what was the normal temperature?
- All inoculated animals showed the same clinical signs ont he same days? were there any individual differences in the showed symptoms?
- line 235 – what do you mean by „immune resilience”?
Table 3. The numbers represent an average value of the infected cattle?
- 3.7. Skin in general everywhere, or histopathological findings were observed only near to the inoculation sites?
Discussion
We should not forget, that the inoculation site, tissue, and the inoculation method, the admistered amount of virus were unnatural. An animal in the field is infected by oral or tracheal way with unknown (probably low) amount of virus. This can alter the onset and progression of the disease. The authors should emphasize and underline it somewhere in the discussion.
lines 405 – 423 – This lines should be part of the introduction. Discussion chapter is for explaining our own data.
lines 443-448 – why the authors selected this virus strain for the study? Should be explained.
Even number of the animals was low, did the authors observed any individual differences between the reaction of cattle to the virus infection?
Spelling:
line 61 – A recent study ……
line 62 - the lung, space mistake
line 96 – in an indigenous ….. (and is not necessary)
line 122 – as controls.
line 179 – for histopathology.
line 186 – testical cells
line 188 – peni-cillin
Round 2
Reviewer 1 Report
Comments and Suggestions for Authors
The authors have revised the manuscript in accordance with my comments. I agree with these corrections and additional information.
As a small note, the commas in Table 2 should also be replaced with periods.